# Self-Adjuvanting Calcium-Phosphate-Coated Microcrystal-Based Vaccines Induce Pyroptosis in Human and Livestock Immune Cells

**DOI:** 10.3390/vaccines11071229

**Published:** 2023-07-11

**Authors:** Yolanda Corripio-Miyar, Clair Lyle MacLeod, Iris Mair, Richard J. Mellanby, Barry D. Moore, Tom N. McNeilly

**Affiliations:** 1Moredun Research Institute, Pentlands Science Park, Penicuik EH26 0PZ, UK; 2Department of Pure and Applied Chemistry, University of Strathclyde, Glasgow G1 1XQ, UK; 3The Roslin Institute, Royal (Dick) School of Veterinary Studies, The University of Edinburgh, Midlothian EH25 9RG, UK; 4Lydia Becker Institute of Immunology and Inflammation, Faculty of Biology, Medicine and Health, The University of Manchester, Manchester M13 9PT, UK

**Keywords:** immune cells, pryroptosis, adjuvant

## Abstract

Successful vaccines require adjuvants able to activate the innate immune system, eliciting antigen-specific immune responses and B-cell-mediated antibody production. However, unwanted secondary effects and the lack of effectiveness of traditional adjuvants has prompted investigation into novel adjuvants in recent years. Protein-coated microcrystals modified with calcium phosphate (CaP-PCMCs) in which vaccine antigens are co-immobilised within amino acid crystals represent one of these promising self-adjuvanting vaccine delivery systems. CaP-PCMCs has been shown to enhance antigen-specific IgG responses in mouse models; however, the exact mechanism of action of these microcrystals is currently unclear. Here, we set out to investigate this mechanism by studying the interaction between CaP-PCMCs and mammalian immune cells in an in vitro system. Incubation of cells with CaP-PCMCs induced rapid pyroptosis of peripheral blood mononuclear cells and monocyte-derived dendritic cells from cattle, sheep and humans, which was accompanied by the release of interleukin-1β and the activation of Caspase-1. We show that this pyroptotic event was cell–CaP-PCMCs contact dependent, and neither soluble calcium nor microcrystals without CaP (soluble PCMCs) induced pyroptosis. Our results corroborate CaP-PCMCs as a promising delivery system for vaccine antigens, showing great potential for subunit vaccines where the enhancement or find tuning of adaptive immunity is required.

## 1. Introduction

Despite its relatively short history, vaccination has resulted in the successful control of several major infectious diseases affecting both humans and livestock species in many parts of the world. Faced with the spectre of untreatable multidrug-resistant infections due to our dependence on antimicrobials and anthelmintics [1], vaccination remains one of the key options for sustainable control of infectious diseases in both farmed animals and humans. While vaccines were initially developed empirically using live attenuated killed pathogens or pathogen fractions, the recombinant DNA revolution in the 1980s enabled the production of many new protein antigens and promised to revolutionise vaccine design, enabling vaccines containing recombinant antigens (subunit vaccines) to be rapidly developed [2,3]. Such recombinant vaccines have many attractive features, including excellent safety profile (as they are non-replicating in vivo and can be obtained in relatively pure form), good stability, scalability and moderate cost of manufacture [4]. However, it was soon discovered that most recombinant antigens are poorly immunogenic, requiring co-administration of adjuvants to exert an appropriate immunological response [5,6]. Furthermore, aluminium salts (Alum), the oldest and most common adjuvants licensed for use in humans, are generally poor at inducing long-lasting immune responses, produce local inflammation at the site of injection, provide limited cell-mediated immune responses, are ineffective for some subunit antigens, and suffer from cold-chain instability issues [7,8,9,10]. Alternative methods have since been developed for presenting “subunit antigens” within more complex immuno-constructs (such as adenoviruses, virus-like particles, and self-amplifying mRNA), but these come with the penalty of increased costs, challenging manufacture and lower stability.

Many different adjuvant systems have been developed, and they can be broadly categorised as either delivery systems or immune potentiators [11,12]. Delivery systems, which are generally particulate, e.g., emulsions, microparticles, immune stimulating complexes (ISCOMS) and liposomes, enhance the exposure of antigen to the immune system or target antigen to specific locations within the body, where uptake by antigen-presenting cells leads to enhanced immunity [13,14]. In contrast, immune potentiators, often purified Pathogen-Associated Molecular Patterns (PAMPs), act directly on immune cells to activate innate immune pathways involved in the induction of adaptive immune responses [15]. Research into immune potentiators has benefitted from an increasing appreciation that innate immune signalling plays a critical role in the initiation, amplification and type of adaptive immune response induced [16]. Indeed, toll-like receptors have been shown to have positive effects during immunisation, with many compounds currently being assessed for their suitability and efficacy as immune adjuvants [17,18,19,20,21]. The use of individual or combinations of PAMPs that drive different immune responses allows for the production of vaccines tailored to the individual pathogens [22,23,24]. Danger-associated molecular patterns (DAMPs) have also been suggested as a possible adjuvant [25]. Alum is perhaps the most well-known DAMP-inducing adjuvant: through a pathway named pyroptosis, a cell death mechanism distinct from apoptosis [26,27,28], it induces inflammasome activation and consequently the secretion of Interleukin (IL)-1β and IL-18 by immune cells [29], alongside the release of DNA from host cells [30]. Principally used as a mechanism for protecting the host against microbial pathogens, pyroptosis has recently been explored as a possible vaccine adjuvant and was shown to delay tumour growth and increase survival in mice immunised with an active Caspase-1 adjuvanted antigen DNA vaccine [31]. Finally, a suitable antigen–adjuvant combination is paramount for eliciting long-lasting immunity and protection [23].

Protein-coated microcrystals (PCMCs) are a novel vaccine delivery method in which vaccine antigens are immobilised in order to enhance antigen stability [32,33]. The substantial cold chain requirement for most commercial vaccines is a recognised limitation of the large-scale production and distribution of vaccines, with more than 50% of available doses being discarded [34,35]. The recent COVID-19 pandemic proved that it is possible to achieve large-scale global production and distribution of vaccines, even if these require cold storage. However, due in part to the short shelf life of most COVID-19 vaccines, wealthy countries stocked millions of surplus doses, which are still being disposed of [36]. PCMCs have been shown to possess the ability to bypass some of these problems, have been demonstrated to be highly stable powder compounds that can be dissolved in the buffer of interest, and have the potential to increase the immunogenicity of certain antigens [32,37,38]. The original PCMC formulation could be further refined by the incorporation of a coating of calcium phosphate (CaP), leading to a more balanced T-helper type 1(Th1)/T-helper type 2 (Th2) immune response when compared to aluminium-adjuvanted formulation [33]. Indeed, CaP has been shown to be a potent natural adjuvant that causes minimal inflammation [9]. Furthermore, the sparingly soluble properties of CaP-PCMCs has been hypothesised to be the underpinning factor for the depot effect observed after immunisation, which is likely to contribute to the immunogenicity and adjuvant properties of the CaP-PCMCs [33]. During the single-step manufacturing process of CaP-PCMCs, antigens can also be co-immobilised with other immune potentiators such as PAMPs, allowing vaccines to be efficiently tailored to specific pathogens.

The potential depot effects of CaP-PCMCs, as well as the ability of co-immobilising antigens with immune potentiators, make this vaccine delivery system attractive for use in future subunit vaccines [33]. However, to fully exploit the potential of this novel delivery system, a better understanding of the mechanism of adjuvancy is required in host species relevant to global vaccination programmes in order to understand the potential synergies and/or antagonistic relationships with other vaccine technologies. Hence, in this study, we set out to investigate the nature of the cellular responses involved in CaP-PCMC interactions with immune cells from humans, and two globally important livestock species, cattle and sheep. Our initial in vitro assays showed that CaP-PCMCs induced a rapid cell death in peripheral blood mononuclear cells. This effect was further investigated using model immune cells, which established that these responses were, in fact, of a reproducible particle–cell contact-mediated inflammatory nature, and could be classified as pyroptotic events. These in vitro studies show the potential of CaP-PCMCs not only to be used as a vehicle for antigens, but also to induce defined and strong immune responses in the host.

## 2. Materials and Methods

### 2.1. Animals and Human Donors

Healthy 8–12-month-old male Holstein–Friesian cattle (approximately 350 kg) and 2–3-year-old female old Texel-cross sheep (approximately 85 kg) were kept at the Moredun Research Institute (MRI) facilities. No animals had a history of being in a grazing pasture in order to exclude accidental infection with helminth parasites, and the animals were considered helminth-naïve. All experiments were carried out under license from the Home Office Guidelines (Licence No. PPL 60/3854) following approval by the Moredun Animal Welfare and Ethical Review body.

Ex vivo experiments with fresh human peripheral blood from healthy donors were approved by the Accredited Medical Regional Ethics Committee (AMREC, reference number 15-HV-013) and with informed consent from all participants.

This study was carried out in compliance with the ARRIVE guidelines (https://arriveguidelines.org, accessed on 2 May 2022), with the entire methodology being carried out in accordance with relevant guidelines and regulations.

### 2.2. Formulation of CaP-PCMCs

The CaP-PCMC formulations were prepared as previously described [33]. Briefly, aqueous mixtures of Ovalbumin (OVA), as well as histidine (His), were precipitated by addition into a 19-fold excess of isopropanol containing dissolved calcium chloride. The resultant suspension contained self-assembled microcrystals comprising an amino-acid core with the protein embedded in a thin surface layer of calcium phosphate. CaP-PCMCs were then isolated by vacuum filtration onto PVDF hydrophilic 0.45 μm filters (Merck Millipore, Hertfordshire, UK) and dried overnight to a dry powder. Protein content and integrity were determined by bicinchoninic protein assay kit (Pierce Biotechnology, Inc., Rockford, IL, USA) and SDS-PAGE.

### 2.3. Isolation of Bovine and Ovine Peripheral Blood Mononuclear Cells (PBMC) 

Blood was collected aseptically by venipuncture into 350 mL blood bags containing 45 mL of Citrate phosphate dextrose-adenine 1 (CPDA-1) stabiliser (Sarstedt, Nümbrecht, Germany) or in sodium heparin vacutainers (Becton Dickinson, Oxford, UK). PBMCs were isolated as previously described [39] using Ficoll-Paque™ PLUS (GE Healthcare Life Sciences, Chicago, IL, USA). Interphase was collected, and following three washes with PBS, PBMCs were re-suspended at 2 × 10^7^ cells/mL in tissue culture media (RPMI-1640 medium supplemented with 10% FBS (Sigma-Aldrich, St. Louis, MO, USA), 50 µM 2-mercaptoethanol, 2 mM L-glutamine, 100 U/mL Penicillin, 100 µg/mL Streptomycin).

### 2.4. Monocyte and Lymphocyte Purification from Cattle PBMCs 

Monocytes, B cells and T cells were purified from PBMCs using magnetic separation. Briefly, PBMCs were resuspended in cold MACS buffer and incubated with CD14 MicroBeads (clone TÜK4, Miltenyi Biotech, Bergisch Gladbach, Germany) for 15 min at 4 °C. Cells were then washed by centrifugation and resuspended in MACS buffer before purification over an LS column. Following elution of CD14 monocytes from the column, cells were washed to eliminate any residual magnetic beads and resuspended in tissue culture media. CD3^+ve^ and CD3^-ve^ populations were further purified by incubating the CD14^-ve^ fraction with a bovine CD3 monoclonal antibody (clone MM1A, Kingfisher Biotech, Inc. Saint Paul, MN, USA), followed by coating with Anti-IgG MicroBeads (Miltenyi Biotech). The CD14^-ve^CD3^+ve^ and CD14^-ve^ CD3^-ve^ fractions were then separated using a magnetic separator as detailed above. The cell purity of each fraction was consistently around 95%. CD14^+ve^, CD14^-ve^CD3^+ve^ and CD14^-ve^ CD3^-ve^ populations were used in the apoptosis studies detailed below.

### 2.5. CaP-PCMCs Induced Apoptosis in Cattle Immune Cells

#### 2.5.1. Detection of CaP-PCMCs Induced Apoptosis 

Apoptosis induced by the incubation of immune cells with CaP-PCMCs was detected with an Annexin V Apoptosis Detection Kit FITC (eBioscience, ThermoFisher, Waltham, MA, United States). In the initial experiment, carried out to determine the onset of apoptosis, 10^6^ PBMCs were incubated with 1.25 mg/mL CaP-PCMCs for 15 min, 30 min or 60 min. This was achieved by adding 50 μL of 5 mg/mL of CaP-PCMC-OVA resuspended in PBS to 150 µL PBMCs in complete tissue culture medium. Subsequently, PBMCs were incubated with two-fold serial dilutions of CaP-PCMCs starting from 1.25 mg/mL in order to determine the concentration at which CaP-PCMCs would cease to induce apoptosis. Finally, to determine whether there were cellular populations in PBMCs that were more sensitive to apoptosis, purified CD14^+ve^, CD14^-ve^CD3^+ve^ or CD14^-ve^CD3^-ve^ cells were incubated with 1.25 mg/mL CaP-PCMC-OVA for 30 min.

After the incubation periods, cells were pelleted by centrifugation and washed with 200 µL of PBS. Cells were washed once more with 1 x Annexin Binding Buffer. Once resuspended in 200 µL of binding buffer, 5 µL of Annexin:FITC was added to the wells and incubated at RT in the dark for 15 min. Finally, cells were washed twice with binding buffer, and 5 µL of Propidium Iodide (PI) was added to the experimental wells prior to analysis in order to distinguish early from late apoptotic cells.

For this and all subsequent flow cytometry assays, a minimum of 50,000 events were acquired using a MACSQuant^®®^ Analyzer 10 (Miltenyi Biotech) and analysed using FlowJo vX Software for Windows 7 (BD Life Sciences, Franklin Lakes, NJ, USA). Cells were gated to eliminate duplets (Figure 1A), single cells were then gated further based on SSC-A vs. FSC-A to eliminate any debris (Figure 1B). Finally, cells were gated into four populations (Figure 1C,D): live (PI negative, Annexin V negative), late apoptotic (PI positive, Annexin V positive), early apoptotic (PI negative, Annexin V positive), and dead (PI positive, Annexin V negative).

#### 2.5.2. Influence of CaCl_2_ and CaP-PCMC Composition on Apoptosis

PBMCs were incubated with microcrystals formed with calcium phosphate coating (CaP-PCMC-OVA), microcrystals without calcium phosphate coating (soluble PCMC-OVA), or with 206.25 µg/mL of dissolved CaCl_2_ (Sigma-Aldrich). The latter provided a concentration of soluble calcium ions equivalent to the maximum that could be released from the CaP-PCMC-OVA particles added to cells. Following the resuspension of CaP-PCMC-OVA and soluble PCMC-OVA in PBS, solutions were spun down at 2000 rpm for 5 min in order to pellet the insoluble fraction of the microcrystals. Supernatants were then collected and used as an extra treatment for comparison with CaP-PCMC-OVA and soluble PCMC-OVA. Cells were incubated with the various treatments for a period of 30 min, after which samples were washed and stained with Annexin V and PI as detailed above.

#### 2.5.3. Cell–CaP-PCMC Interaction Studies

PBMCs were seeded into a HTS Transwell-96 Permeable Support with a 0.4 μm pore polycarbonate membrane (Sigma-Aldrich), as follows. A total of 10^6^ PBMCs per well (the top well contained 5 × 10^5^ PBMCs, as the volume was lower than that of the bottom well) were incubated alongside CaP-PCMC-OVA or separated by the transwell. Different combinations were used, i.e., PBMCs and CaP-PCMC-OVA in the same well, PBMCs in the top well and CaP-PCMC-OVA in the bottom well, and vice versa, in order to determine if contact with the crystals or if material released by the crystals was the cause of apoptosis. After 30 min incubation, cells were harvested from the top or bottom well and kept separate during staining with Annexin V and PI, as detailed above.

### 2.6. Generation of Monocyte-Derived Dendritic Cells (MoDCs)

Vaccine antigens must be phagocytosed and processed before being presented to the immune system. As professional antigen-presenting cells, dendritic cells are a cell population critical during these initial moments following vaccination, initiating and directing the immune responses. Hence, monocyte-derived dendritic cells (MoDCs) could be a very useful in vitro tool for investigating how CaP-PCMCs and antigen-presenting cells could interact following vaccination in vivo. Consequently, we produced MoDCs from cattle, sheep and human-purified CD14^+ve^ monocytes to further our in vitro studies.

Cattle and sheep MoDCs were generated as previously described [39]. Briefly, purified cattle CD14^+ve^ monocytes were seeded in 6-well plates at a concentration of 10^6^ cells/mL and cultured in the presence of 50 µL of Bovine or Ovine DC Growth Kit (BioRad, Hertfordshire, UK) per mL of media at 37 °C, 5% CO_2_ for 3 days for cattle MoDCs, while sheep MoDCs required 5 days’ incubation, with cytokines and media being replenished on day 3.

For human MoDCs, PBMCs were isolated from peripheral blood from healthy donors by a density gradient using Ficoll-Plaque (Sigma-Aldrich, Merck Life Science UK Limited, Dorset, UK). Monocytes were magnetically purified as detailed above for cattle and sheep, and subsequently differentiated into MoDCs using a human MoDC Generation Toolbox according to manufacturer’s instructions (Miltenyi). In brief, CD14^+ve^ monocytes were re-suspended at 10^6^ cells/mL in MoDC Differentiation Medium and incubated at 37 °C, 5% CO_2_ and >95% humidity for 3 days. On day 3, equal amounts of fresh MoDC Differentiation Medium were added to the culture and cells were incubated for another 4 days.

### 2.7. Caspase-1 Activation Assay

Caspase-1 activation by CaP-PCMC-OVA was determined in cattle, sheep and human MoDCs using an FAM-FLICA^®®^ Caspase-1 Assay Kit (BioRad). The FLICA reagent FAM-YVAD-FMK enters each cell and irreversibly binds to activated Caspase-1, emitting a green fluorescence, which can be detected by flow cytometry [40]. Briefly, MoDCs were harvested and adjusted to a concentration of 10^6^ cells/mL, and then incubated in the presence or absence of 1.25 mg/mL CaP-PCMC-OVA (final concentration) for 2 h, 6 h and 24 h. Samples processed for Caspase-1 were incubated with 5 µL of 30 × FLICA solution alongside the CaP-PCMC-OVA or the same volume of media (control). After the incubation period, media were collected for cytokine analysis and the cells were harvested. Cells were washed twice with apoptosis buffer, re-suspended in the buffer and PI, and analysed by flow cytometry. Apoptosis during this time course was determined in parallel using Annexin V as detailed above.

### 2.8. Quantification of Cytokines by ELISA

Interleukin (IL)-1β release by bovine and ovine MoDCs was quantified by ELISA using high-binding-capacity ELISA plates (Immunolon™ 2HB 96-well microtiter plates, ThermoFisher). All washing steps were carried out 6 × with PBS + 0.05% Tween 20 (PBST), using an automatic washer (Thermo Scientific Wellwash™ Versa, ThermoFisher). Initially, plates were incubated with capture antibody (polyclonal anti-bovine IL-1β, BioRad) overnight at 4 °C. Antibody was then washed with PBST and plates were blocked for 1 h at RT with PBS + 3% Bovine Serum Albumin (BSA, Sigma) to avoid nonspecific binding. Samples and standards (recombinant bovine IL-1β ranging from 40 ng/mL to 50 pg/mL, BioRad) were incubated in duplicate for 1 h at RT, and then the IL-1β polyclonal detection antibody (BioRad) was added for a further hour. After washing with PBST, streptavidin-HRP (Sigma) was added to all wells for 45 min. IL-1β was then detected by the addition of SureBlue TMB substrate (Insight Biotechnology, Wembley, UK), and 2.5 M H_2_SO_4_ was added to stop the reaction.

For human IL-1β, a Human IL-1 beta ELISA Ready-SET-go kit (2nd generation, Affymetrix, Santa Clara, CA, USA) was used following the manufacturer’s instructions. Briefly, Corning Costar 9018 ELISA plates were coated with capture antibody and incubated overnight at 4 °C. Plates were then washed and blocked for 1 h with 1 xELISA/ELISPOT diluent. Following a washing step, samples or standards were added to the plate for 2 h, washed and detection antibody was added to all wells for 1 h. After a final washing step, 100 µL of TMB solution was added and the reaction stopped with 50 µL of stop solution.

In all cases, absorbance values were measured as optical density at 450 nm using a Sunrise™ microplate reader (Tecan, Männedorf, Switzerland). All values were blank corrected, and concentrations were determined from the standard curves.

### 2.9. Statistical Analysis

Statistical analysis for MoDC experiments was performed using Two-Way ANOVA (with time and treatment as factors) and Tukey’s post hoc test. Non-normally distributed data were log-transformed prior to analysis to achieve normality as assessed with the Anderson–Darling normality test. All analyses were carried out using the GraphPad Prism version 8.0.0 statistical package for Windows, (GraphPad Software, San Diego, CA, USA), with *p* < 0.05 being considered significant.

## 3. Results

### 3.1. In Vitro Effect of CaP-PCMCs on Cattle PBMC-Apoptosis Studies 

#### 3.1.1. Induction of Cell Death by CaP-PCMCs

In order to determine the effect of CaP-PCMCs containing OVA as a model antigen on immune cells in vitro, a series of experiments was carried out. Initially, cattle PBMCs were incubated with 1.25 mg/mL of CaP-PCMC-OVA re-suspended in PBS for periods of 15, 30 and 60 min in 96-well plates. Following the harvesting of cells by centrifugation, cell death was determined by staining with Annexin V and PI. Flow cytometry analysis revealed that even with incubation periods as short as 15 min, cell death was quickly induced, with 55.5% early apoptotic, 37.6% late apoptotic, 0.35% dead and only 6.57% live cells (see Figure 2). Longer incubation times did not increase the level of cell death in PBMCs.

#### 3.1.2. Dose Dependence Studies

We then set out to determine whether the level of cell death induced by CaP-PCMCs was dose dependent (Figure 3). Cattle PBMCs were incubated with a 1:2 serial dilution of microcrystals, starting with a concentration of 1.25 mg/mL for a period of 30 min. The highest concentration resulted in a combined early and late apoptosis of 81.6%, which decreased in a dose-dependent manner to around 22.5% when incubating cells 0.02 mg/mL of CaP-PCMC-OVA, after which the level of cell death plateaued, and was similar to that seen in untreated control cells.

#### 3.1.3. Levels of CaP-PCMCs Induced Cell Death in White Blood Cell Populations

Since PBMCs are a combination of different white blood cells, including T cells, B cells, NK cells and monocytes, we wanted to establish whether any of these populations were more sensitive to cell death induced by CaP-PCMCs. Following the positive selection of monocytes, the negative population was then split into CD3^+ve^ and CD3^-ve^, resulting in three distinct populations: CD14^+ve^ monocytes, CD3^+ve^ T cells (including cytotoxic and helper T cells and CD3^+ve^ NK cell subsets), and CD3^-ve^ T cells (CD3^-ve^ NK cells and B cells). Subsequent incubation of each of these cell populations with CaP-PCMC-OVA for a period of 30 min demonstrated that cell death was similar for all three populations (Figure 4), and no specific white blood cell populations were more sensitive to the action of CaP-PCMC-OVA than others.

CaP-PCMCs were able to induce apoptosis in vitro in white blood cells in a dose-dependant manner without targeting specific cell types present in the PBMCs.

### 3.2. Investigating Factors That Influence Cell Death: Effects of Composition and Contact with CaP-PCMCs on Apoptosis

#### 3.2.1. Effects of the Composition of CaP-PCMCs on Cell Death

CaP-PCMCs contain high levels of calcium ions as part of their composition; since excess calcium (Ca^2+^) can alter cellular function, leading to apoptosis [41], we wanted to investigate whether this could be the cause of the cell death observed in our experiments. Initially, we tested both the composition of the microcrystals and the free calcium ions—which they were able to provide to the culture—as possible cell-death-inducing factors. The calcium phosphate that coats the CaP-PCMC-OVA causes the crystals to be sparingly soluble, and consequently we produced “soluble PCMC-OVA” without calcium phosphate coating. Due to the manufacturing process, CaP-PCMC-OVA particles may contain a small amount of excess calcium chloride on their surface. In addition, although the calcium phosphate layer on CaP-PCMC-OVA particles is sparingly soluble, it will partially dissolve in aqueous media, also releasing calcium ions [42]. To test whether these ions could be contributing to the observed cell death, we introduced the maximum concentration of calcium ions that could be released on full particle dissolution into the cell medium, which was equivalent to 825 µg/mL of CaCl_2_ per 5 mg/mL CaP-PCMC-OVA (206.25 μg/mL final concentration). Consequently, we incubated cattle PBMCs with soluble PCMC-OVA, CaP-PCMC-OVA, supernatants from both PCMC formulations obtained by centrifugation, or the equivalent concentration of CaCl_2_ present in the CaP-PCMC-OVA (Figure 5). Staining with Annexin V revealed that the only treatment in which cell death could be detected was that in which the cells were incubated with the CaP-PCMC-OVA (Figure 5) particles present. None of the individual particle components, namely, ovalbumin, glutamine, phosphate, or calcium ions, caused cell death when dissolved in the cell media, showing that only those cells with CaP-PCMCs in the culture were driven to apoptosis.

#### 3.2.2. Effect of Cell-to-CaP-PCMC Contact on Cell Death

Finally, we investigated whether this cell death was driven by direct cellular contact with the microcrystals. For this purpose, we carried out a series of experiments where PBMCs and CaP-PCMC-OVA were separated by a porous 0.4 μm membrane (CaP-PCMC-OVAs are smaller than 0.45 μm) within the tissue culture well, creating upper and lower chambers within the well, allowing passage of soluble components of CaP-PCMC-OVA but preventing contact between cells and CaP-PCMC-OVA. For this assay, cells were incubated in the upper or lower chamber of the tissue culture well, with CaP-PCMC-OVA in the opposite chamber. Positive cell death controls included cells incubated with CaP-PCMC-OVA in the same chamber. Cells were also cultured with cells plus CaP-PCMC-OVA in the opposite chamber to account for any secretion from dead or dying cells, which could in turn induce cell death in neighbouring cells. After 30 min incubation and staining with Annexin V, we observed that only those cells incubated with CaP-PCMC-OVA in the same chamber were dead or dying (Figure 6A,B).

Apoptosis was not induced by the components of the CaP-PCMCs, and only when cells shared the same space with microcrystals.

### 3.3. CaP-PCMCs Induced Apoptosis in Model Antigen-Presenting Cells in Three Mammalian Species

Following the injection of a vaccine, professional antigen-presenting cells such as dendritic cells are key during initial protein antigen uptake. Thus, we set out to further investigate the mechanism of cell death induced by CaP-PCMC-OVA and its effect on antigen-presenting cells using model innate immune cells, namely Monocyte-derived Dendritic Cells (MoDCs), across three different mammalian species—cattle, sheep and humans. Pyroptosis is a pathway to cell death that is triggered by Caspase-1, leading to the activation of pro-inflammatory cytokines IL-1β and IL-18. During this process, the cells swell up, leading to cell lysis and consequent release of inflammatory cellular content [26,27]. Apoptosis, on the other hand, is not inflammatory in nature [27]. Consequently, and in order to investigate whether the induced cell death mechanism observed up until this point was a pyroptotic or an apoptotic event, secretion of IL-1β and Caspase-1 activation was also quantified during these experiments. The results are shown in Figure 7. As observed in cattle PBMCs, Annexin V staining revealed that the cell death of the bovine MoDCs occurred at all time points investigated, and in similar levels in all species studied (Figure 7A–C). A significant increase in IL-1β secretion by bovine MoDCs incubated with CaP-PCMC-OVA was observed when compared with their corresponding control incubated with an equivalent volume of PBS (untreated) after 6 h and 24 h (F(1, 1) = 588.8, *p* = 0.0262; Tukey’s post hoc test, *p* = 0.0270 for 6 h and *p* = 0.0057 for 24 h when compared to untreated; Figure 7D). CaP-PCMCs also induced a steady increase in Caspase-1 activation over time when compared to untreated (F(1,1) = 472.3, *p* = 0.0293; Tukey’s post hoc test, *p* = 0.0159 for 6 h and *p* = 0.0466 for 24 h; Figure 7G).

Similar results were observed for sheep, with a significant increase in IL-1β secretion observed in cells incubated with CaP-PCMCs over time (F(1, 3) = 439.5, *p* = 0.0002; Figure 7E), starting at 6 h (Tukey’s post hoc test, *p* = 0.0021 compared to untreated) and peaking at 24 h (Tukey’s post hoc test, *p* = 0.0002 compared to untreated). Caspase-1 activation was significantly increased in CaP-PCMCs (F(1, 1) = 378.0, *p* = 0.0327; Figure 7H) from as early as 2 h post incubation (Tukey’s post hoc test, *p* = 0.0138), producing increasing levels of Caspase-1 as cells were exposed to CaP-PCMCs for longer periods (Tukey’s post hoc tests, 6 h, *p* = 0.0129; 24 h, *p* = 0.0106; Figure 7H).

In human MoDCs (Figure 7C,F,I), the increase in IL-1β secretion following incubation with CaP-PCMC-OVA did not reach statistical significance (F(1, 1) = 92.62, *p* = 0.0659; Figure 7F). However, we did observe a significant increase between the untreated controls and cells incubated with CaP-PCMC-OVA for Caspase-1 (F(1, 2) = 360, *p* = 0.0028; Figure 7I). This significance was observed at both time points, following incubation with CaP-PCMC-OVA after 6 h (Tukey’s post hoc test, *p* = 0.006; Figure 7I) and at 24 h (Tukey’s post hoc test, *p* < 0.0001; Figure 7I), when Caspase-1 levels reached their peak.

The cell death observed when PBMCs and MoDCs were incubated alongside CaP-PCMCs was in fact a pyroptotic event, as demonstrated by the activation of Caspase-1 and the high levels of secretion of IL-1β.

## 4. Discussion

In recent years, vaccine development has been focused on the design of safer and simpler vaccines. However, while vaccines based on recombinant proteins or peptides have generally improved the safety profiles, they are frequently less immunogenic and require the tailored use of adjuvants to enhance the specific immune responses. Understanding the mode of action of these adjuvants is key, as different adjuvants will elicit a variety of immune responses [43,44]. Adjuvants such as Alum, which is commonly used in veterinary species and is the only adjuvant routinely used in human vaccines, have been shown to act as a depot, with antigens being slowly released and phagocytosed by antigen-presenting cells. This system has been postulated to be more efficient than pinocytosis [45], thereby enhancing immune response [46]. However, Alum can also have adverse effects in humans, with local reactions, lack of effectiveness (particularly for recombinant antigens), poor induction of cell-mediated immune responses, or promotion of IgE-mediated allergic reactions having been reported [47]. In veterinary species, and particularly in sheep, Alum can also be problematic, with large granulomas developing following immunisation [48]. In cats, this local inflammation and granuloma formation were associated with a high incidence of sarcomas during the 1980s [49]. All this, combined with the need to drive more specific immune responses, has led to the exploration of novel adjuvants. Here, we investigated the adjuvancy properties of calcium-phosphate-protein-coated microcrystals (CaP-PCMCs), a novel adjuvant/delivery system with the potential to be used in veterinary vaccines. PCMCs have previously been shown to be thermally stable [32], and are able to deliver a balanced Th1/Th2 immune response in mice [33]. They benefit from containing calcium phosphate, a natural component of the body that is not only very well tolerated, but has also been shown to be an excellent carrier for nucleic acids or drugs in many biological systems [50]. Furthermore, when used as an adjuvant, CaP is able to elicit high IgG2a and neutralising antibody titres in mice, as well as an increased IgE response [9]. Even though CaP-PCMCs as novel adjuvant systems have been proven to be efficient and effective in murine models [51,52], prior to the work presented here, the mechanism of adjuvancy of CaP-PCMCs was still unknown.

In order to address this, we set out to investigate how CaP-PCMCs interact with the immune system in a series of in vitro cellular assays. In our initial experiments, we focused on cattle cells, and found that when bovine PBMCs were incubated with CaP-PCMCs, co-formulated with ovalbumin as a model antigen, for a period as short as 15 min, over 85% of the cells underwent dose-dependent cell death, affecting all cell populations present in PBMCs to a similar level. Excess calcium ions in cells have been shown to induce apoptosis [53], and theoretically, the sparingly soluble calcium phosphate in CaP-PCMC-OVA, if dissolved, could produce a higher concentration in our in vitro assays than physiological levels [54]. However, we were able to show that if soluble CaCl_2_ was added, this did not induce cell death at the maximum concentration of calcium ions that could be obtained from the microcrystals. Rather, we demonstrated that this cell death was exclusively due to direct contact with the CaP-PCMC-OVA and was unaffected by excess calcium or any other secretions from dead cells undergoing cellular apoptosis. Apoptosis can be an important part of the adjuvancy mechanism of some adjuvants. This is the case of MF56, where following intramuscular immunisation, apoptotic or necrotic macrophage fragments are taken up by resident dendritic cells which subsequently presented antigens to naïve T cells in a more efficient manner [55].

Dendritic cells are key in initiating early responses to vaccination, as they are involved in antigen presentation and processing [56]. The ability to generate dendritic cells derived from blood monocytes for in vitro studies allowed us to investigate the interactions of these cells with CaP-PCMCs further. Consequently, we set out to explore the induction of cell death by CaP-PCMCs by investigating the kinetics of activation of Caspase-1 and release of IL-1β following incubation of MoDCs with CaP-PCMC-OVA. We found that with as little as 6 h of incubation with the microcrystals, Caspase-1 was activated, followed by the release of IL-1β into the culture media, an event common in all three species studied, i.e., cattle, sheep and humans. This cell death, combined with Caspase-1 activation and the secretion of IL-1β, indicates that the microcrystals induced pyroptosis, a form of cell death morphologically and mechanistically different from apoptosis [26,27]. During apoptosis, cells are packed in apoptotic bodies, which are taken up by professional phagocytes, such as macrophages and dendritic cells, and no inflammatory response is triggered [27]. However, during pyroptosis, a controlled loss of membrane integrity occurs with the release of the cellular contents, including myriad cytokines, chemokines, pyroptotic bodies and viral or bacterial fragments [26,27,57]. Eventually, an inflammatory response is triggered, and immune cells infiltrate into the site of infection or the tissue where damage has occurred [27,58,59]. Initially identified as a novel cell death following infection with *Salmonella* and *Shigella* sp. [60,61], pyroptosis has been shown to be key in the clearance of pathogens released by infected cells, triggering neutrophil-mediated killing by phagocytosis, degranulation, and the release of Reactive Oxygen Species (ROS), as well as neutrophil nets [62]. In addition to this, danger-associated molecular patterns (DAMPs) and alarmins are also released during pyroptotic events, thus maintaining the inflammatory processes [63]. This is uniquely dependent on Caspase-1 activation, through which the mature forms of IL-1β and IL-18 are cleaved and released. In our experiments, we detected the activation of Caspase-1 and the consequent release of IL-1β, thus indicating that cell death was pyroptotic in nature. The induction of pyropotosis is not unique to the CaP-PCMC adjuvant system. DNA vaccines are potent priming systems, and are able to produce full-length immunogens in vivo and elicit a balanced antibody and T cell immunity. In addition to this, recent work has highlighted that following immunisation with a DNA vaccine, the pathogen recognition receptor absent in melanoma 2 (Aim2) plays a key role in directing pyroptotic cell death, thereby contributing to the adjuvant effects of this vaccine [64].

The adjuvant activity of IL1-β has been well documented, and its secretion has been shown to be sufficient to support T cell responses to antigens without requiring other adjuvants [7,65]. Alongside IL-6 and IL-23, IL1-β is able to polarise the naïve T helper cells towards a Th17 phenotype [66], so adjuvants that are able to induce the secretion of IL1-β by immune cells could be beneficial for improving the efficacy of vaccines against those pathogens requiring such responses [63]. This is the case for *Mycobacterium tuberculosis* (Mtb), where a mixed Th1/Th17 response is required to acquire immunity against the bacteria [67,68]. Recently, the combined use of chitosan and MPL as adjuvants during the design of a novel vaccine against Mtb was able to induce the release of Th17 polarising cytokines IL1-β and IL-23, conferring a mixed Th1/Th17 mucosal response and consequent protection in challenged mice [69]. Other particulate adjuvants, such as biodegradable poly(lactide-co-glycolide) (PLG) and polystyrene microparticles have also been shown to activate the NALP3 inflammasome increasing the secretion of IL-1β by dendritic cells [70]. Interestingly, Alum, which is a Th2-polarising adjuvant, is also capable of activating Caspase-1 via the activation of NLRP3 and secretion of the pro-inflammatory cytokines IL-1β and IL-18 by macrophages [7,46,71], although it is unclear whether inflammasome activation is essential for Alum-mediated adjuvancy [7,46,72,73].

## 5. Conclusions

We showed here that the mode of action of adjuvancy in CaP-PCMCs is driven by a pyroptotic event, which, by means of IL-1β, could support T cell priming during vaccination and the induction of a Th1 and/or Th17 phenotype. CaP-PCMCs have recently been shown to induce a strong Th2 immunity in a mouse model that can be further boosted by priming with MF59 [51]. Our results, alongside this study, suggest that it may be possible to manipulate responses to induce a balance Th1/Th2 response when using CaP-PCMCs. Indeed, this balanced Th1/Th2 response is desirable in many disease systems [74]. Recent work has shown that this is possible, and that a mixed Th1/Th2 response can be achieved using a squalene-based oil-in-water emulsion adjuvant [75]. A key consideration is that Caspase-1 is able to cleave and inactivate IL-33 [63,76,77], a cytokine involved in T helper 2 differentiation [78]. However, due to the lack of veterinary reagents, we were not able to determine whether IL-33 was inactivated following incubation with CaP-PCMCs, which could potentially impair induction of Th2 responses when using CaP-PCMCs as adjuvants. Work is now required to determine the type of adaptive immune response induced by CaP-PCMCs in livestock and humans and compare this with commercial adjuvants which generally polarise more towards Th2 [75,79]. Furthermore, as CaP-PCMCs allow the incorporation of multiple molecules within the same microcrystal, it is important to determine whether responses to specific antigens can be further manipulated by the incorporation of PAMPs or other immunostimulants into the microcrystals. This may allow the fine tuning of vaccines to generate the most appropriate immune responses for the specific pathogen(s) targeted.

## Figures and Tables

**Figure 1 vaccines-11-01229-f001:**
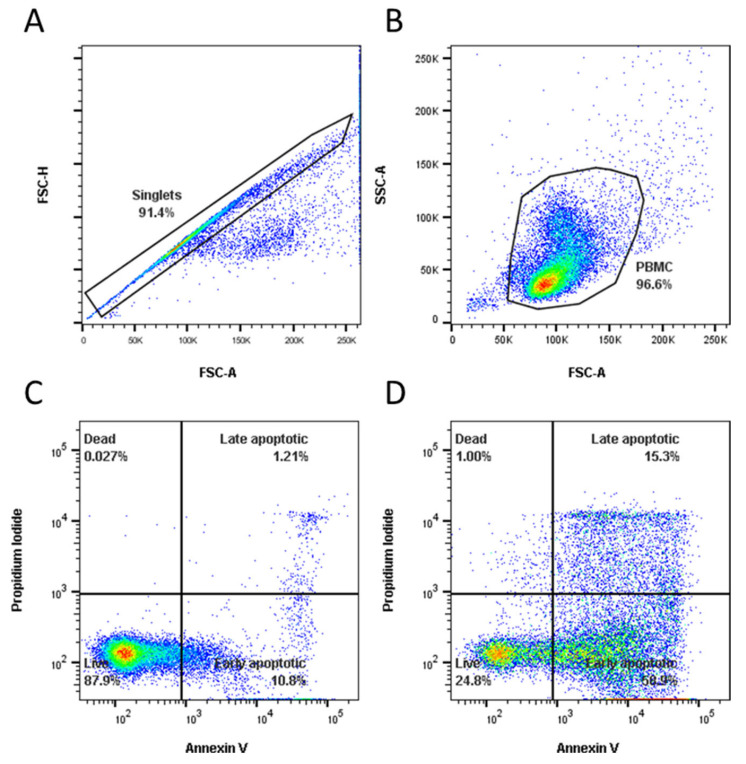
Gating strategy for samples stained with Annexin V Apoptosis Detection Kit FITC. The cells were gated to eliminate doublets (**A**), and then PBMCs were gated (**B**). Samples incubated with media (**C**) or CaP-PCMC-OVA (**D**) were then gated into four populations following the manufacturer’s instructions, as defined by the staining of Annexin V and/or PI: live (double negative), dead (PI positive), early apoptotic (Annexin V positive) or late apoptotic (double positive).

**Figure 2 vaccines-11-01229-f002:**
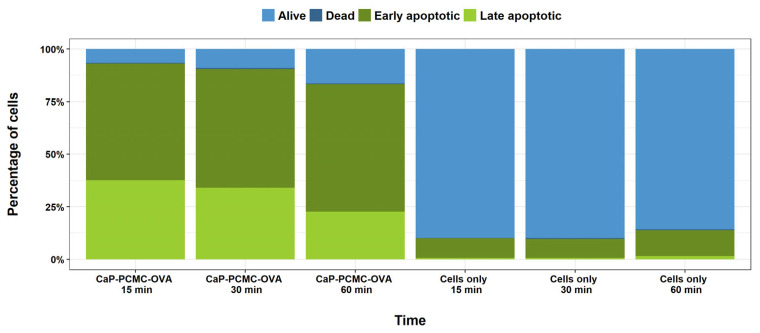
Induced cell death measurements by Annexin V/PI staining in cattle peripheral mononuclear cells following incubation with CaP-PCMC-OVA. Cells were incubated with or without CaP-PCMC-OVA for 15, 30 and 60 min. Following harvesting, cell death was determined by Annexin V/PI apoptosis assay. Single cells were gated into four populations as detailed in Figure 1: live (light blue), dead (dark blue), early apoptotic (dark green) or late apoptotic (light green).

**Figure 3 vaccines-11-01229-f003:**
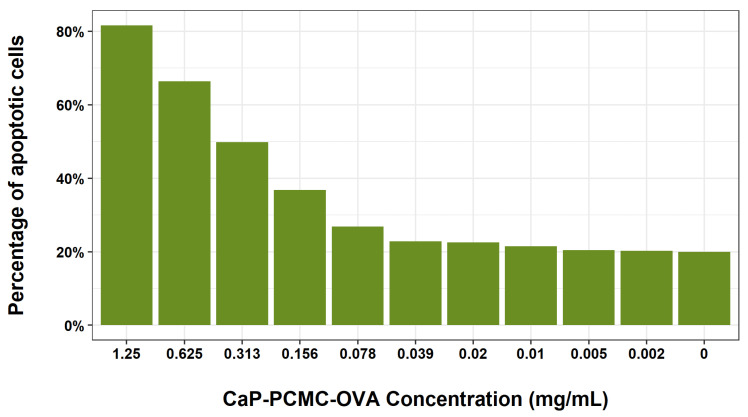
Dose response of cell death induced by Cap-PCMC-OVA. Cattle PBMCs were incubated with a serial 1:2 dilution of CaP-PCMC-OVA starting at 1.25 mg/mL for a period of 30 min. Untreated control cells were also included. Following incubation for 30 min, cell death was determined by Annexin V/PI apoptosis assay. The gating strategy was carried out as detailed in Figure 1, and the percentage of apoptotic cells (Y axis) was determined as the combined percentage of late and early apoptosis.

**Figure 4 vaccines-11-01229-f004:**
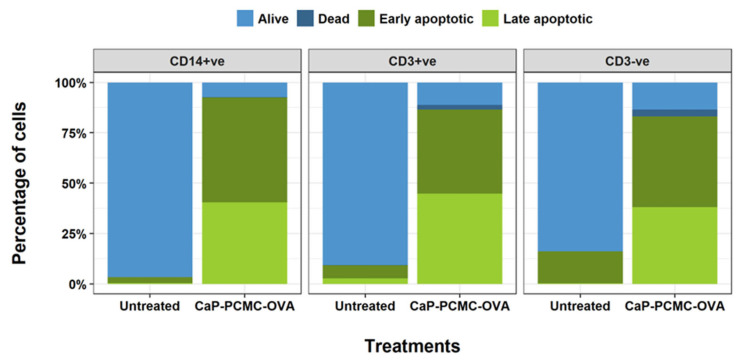
Levels of induced cell death in different cattle PBMC populations induced by CaP-PCMCs. CD14+ve monocytes, CD14-ve CD3+ve T cells, and CD14-ve CD3-ve T cells were separated from PBMCs by MACS and incubated with 5 mg/mL of CaP-PCMC-OVA or in media only (Untreated). Annexin V/PI staining was analysed as detailed in Figure 1. Following incubation for 30 min, cell death was determined by Annexin V/PI apoptosis assay. Single cells were gated into four populations, as detailed in Figure 1: live (light blue), dead (dark blue), early apoptotic (dark green) or late apoptotic (light green).

**Figure 5 vaccines-11-01229-f005:**
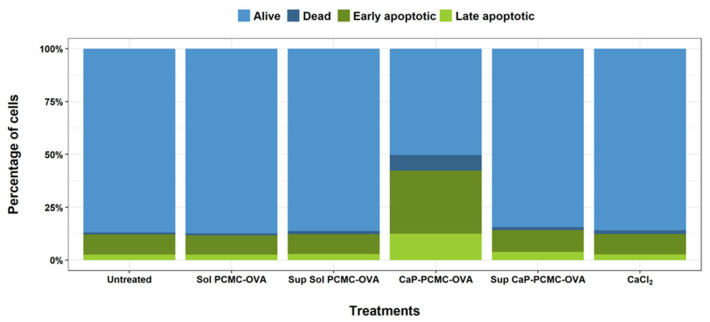
Comparison of induced cell death induced by soluble PCMC components and CaP-PCMCs. Cattle PBMCs were incubated with media only (Untreated), 1.25 mg/mL of soluble PCMCs (Sol PCMC-OVA), 1.25 mg/mL of CaP-PCMC-OVA, supernatants from soluble PCMC-OVA (Sup Sol PCMC-OVA), supernatants from CaP-PCMC-OVA (Sup CaP-PCMC-OVA), or CaCl_2_ to an equivalent concentration of Ca^2+^ found in CaP-PCMC-OVA. Following incubation for 30 min, cell death was determined by Annexin V/PI apoptosis assay.

**Figure 6 vaccines-11-01229-f006:**
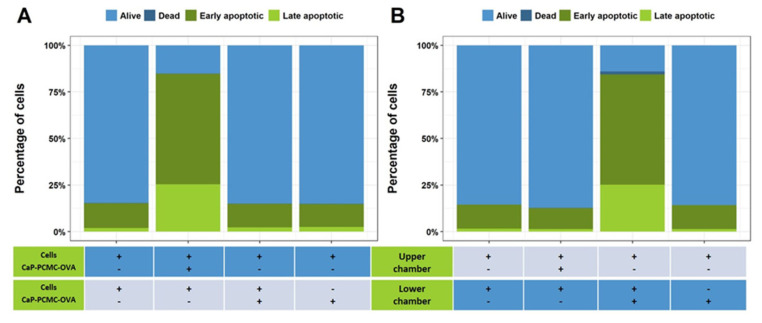
CaP-PCMC-induced cell death is cell-contact dependent. PBMCs were incubated in triplicate in the upper (**A**) or lower (**B**) chamber of a 96-well plate separated by a porous 0.4 μm membrane with (+) or without (-) CaP-PCMC-OVA for 30 min. Cell death was then measured in each separate chamber/combination using an Annexin V/PI apoptosis assay. The cell death results shown are those from the upper chamber (**A**) and lower chamber (**B**).

**Figure 7 vaccines-11-01229-f007:**
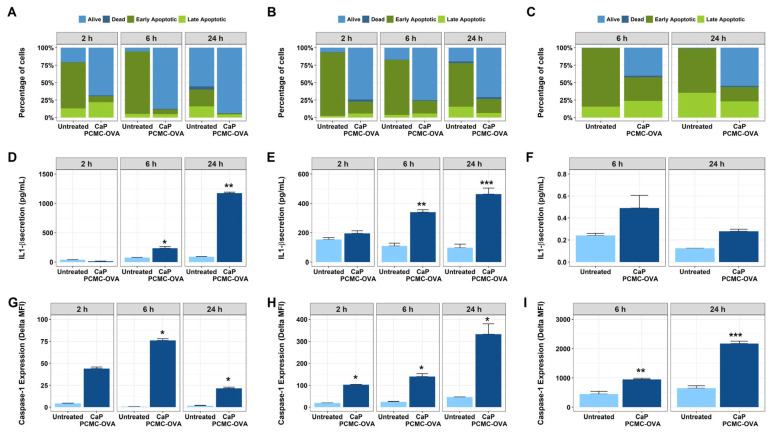
CaP-PCMCs drives pyroptosis in cattle, sheep and human monocyte-derived dendritic cells. Apoptosis, as defined by Annexin V/PI staining (**A**–**C**), IL-1β secretion (**D**–**F**) and Caspase-1 activity (**G**–**I**), was analysed in monocyte-derived dendritic cells after 2 h, 6 h or 24 h incubation with CaP-PCMCs or vehicle alone (untreated) in three mammalian species: cattle (**A**,**D**,**G**), sheep (**B**,**E**,**H**) and human (**C**,**F**,**I**). Annexin V/PI: staining was analysed as detailed in Figure 1. IL-1β secretion: supernatants from MoDCs incubated with CaP-PCMCs were collected, and IL-1β secretion, expressed in pg/mL, was determined by capture ELISA. In parallel experiments, Caspase-1 activity was determined by the irreversible binding of FAM-FLICA peptide, FAM-YVAD-FMK, to active Caspase-1. Intensity of the green fluorescence from bound FAM probe is expressed as the corrected median fluorescence intensity (MFI). Results are shown as mean values with error bars indicating ± SE from three individuals per species. * Denote statistical significance at *p* < 0.05, ** *p* < 0.01 and *** *p* < 0.001.

## Data Availability

All data used in this manuscript will be made available online upon final acceptance of the manuscript.

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
