# Peer review of "Self-Adjuvanting Calcium-Phosphate-Coated Microcrystal-Based Vaccines Induce Pyroptosis in Human and Livestock Immune Cells"

_vaccines, 2023, doi:10.3390/vaccines11071229_

Round 1

Reviewer 1 Report

The present manuscript by Corripio-Miyar et al aims at unraveling the mechanisms of protein coated microcrystals by assaying cell death, Il-1beta and caspase 1 secretion in peripheral blood mononuclear cells and monocyte-derived dendritic cells from cattle, sheep and humans. This aim is sound and important. The experimental set up is adequate and the experiments well performed. However pyroptosi is just one possible programmed inflammatory cell death and tis aspect could be further delt with.

Major issues

1. Additional evidence of pyroptosis could be presented as morphology, Il-18 secretion , gasdermnin detection in order to rule out other forms of programmed cells death as necroptosis.

2. Inhibitors of pyroptosis as could be used to further dissect the mechanisms. 

Minor issues

1. Why were not IL-33. Il-6 and other relevant cytokines  assayed in human cells?p 14 line 550

2. Does IL-1 only promotes TH1 responses? p 14 line 540

Author Response

Major issues

We agree with the reviewer on their comments regarding the expansion of the studies on pyroptosis. Below in italics are our detailed responses to the comments.

 Comment 1. Additional evidence of pyroptosis could be presented as morphology, Il-18 secretion , gasdermnin detection in order to rule out other forms of programmed cells death as necroptosis.

            RESPONSE: Unfortunately, we could not perform cell imaging assays during our in vitro work as the cells were obscured by the CaP-PCMC in the culture. However, using our SSC and FSC flow cytometry plots, were able to see how the CaP-PCMC had induced a change in the size and complexity of the cells indicating cell swelling characteristic of pyroptosis.

Indeed, we agree that secretion of IL18 and activation of gasdermin-D would be interesting markers of pyroptosis to investigate. However, we believe that in this work we have proven that the cell dead induced by the CaP-PCMC is pyroptotic and not necrotic by showing the activation of Caspase-1 and secretion of IL-1β, both of which are clear markers of pyroptosis not of necrosis (See the recent review from Ketelut-Carneiro& Fitzgerald (2022) https://doi.org/10.1016/j.jmb.2021.167378.).

We thank the reviewer for pointing out these additional markers, we agree that it would be interesting to determine if this pyroptosis is GSDMD or GSDME dependent and consequently we will consider these investigations when we carry out more in-depth studies of the pyroptotic mechanism of CaP-PCMC.

Comment 2. Inhibitors of pyroptosis as could be used to further dissect the mechanisms. 

RESPONSE: We agree that using pyroptosis inhibitors in our in vitro assays would be interesting, and we will consider these assays as part of a follow up study as mentioned in the reviewer’s previous comment.

Minor issues 

  1. Why were not IL-33. Il-6 and other relevant cytokines assayed in human cells? p14 line 550

RESPONSE: We agree with the reviewer, these cytokines would be interesting to investigate in human cells. Unfortunately, we did not have enough supernatant available to carry out the assays. Consequently, we decided to keep the cytokine secretion data consistent for all 3 species studied.

 Does IL-1 only promotes TH1 responses? p 14 line 540

RESPONSE: IL-1β is able to promote other T helper subsets such as Th17 as mentioned in line 541 of the original manuscript.

Reviewer 2 Report

Major comments

Introduction

-        L40-46: add appropriate references.

-        L58-67: add appropriate references.

-        L103-108: add appropriate references. 

Materials and Methods

-        L119-123: provide details about the experimental animals (e.g., sex, mean body weight)

-        L141: describe the method of blood sampling  

-        L238-249: add appropriate references.

Minor comments

-        L58-61: Delivery systems, which are generally particulate, e.g., emulsions, microparticles, immune-stimulating complexes (ISCOMS) and liposomes, enhance the exposure of antigens to the immune system or target antigens to specific locations within the body, where uptake by antigen-presenting cells …

-        L71: Danger-associated molecular

-        L81:.. to elicit a long-lasting immunity

-        L82: .. Protein-coated microcrystals (PCMC) are a novel..

-        L87: .. have large-scale global production

-        L99: .. contribute to the immunogenicity and adjuvant property

-        L113-114: .. responses were, in fact, a reproducible  113 particle-cell contact-mediated inflammatory nature,..

-        L127: … and with informed consent from all participants..

-        L216: . Monocyte-Derived

-        L218: .. antigen-presenting cells

-        L223: .. human-purified CD14+ve

-        L256: .. and plates were blocked

-        L265: .. used following the manufacturer’s instructions

-        L269: .. was added to all

-        L273: .. were determined from

-        L275: .. was performed

-        L316: .. Following the positive selection

-        L337: .. CaP-PCMC contains high levels

-        L341: .. death-inducing factors

-        L390: .. antigen-presenting cells

-        L455: .. of cell-mediated

-        L490: .. in antigen presentation and processing

Author Response

We would like to thank Reviewer #2 for their comments to improve our manuscript.  The reponses to all the individual comments are below in italic.

 Comments and Suggestions for Authors

Major comments

Introduction

-        L40-46: add appropriate references.

-        L58-67: add appropriate references.

-        L103-108: add appropriate references. 

            All references have been added 

Materials and Methods

-        L119-123: provide details about the experimental animals (e.g., sex, mean body weight)        Sex, age and body weight added

-        L141: describe the method of blood sampling  

            Blood sampling method (venipuncture) added

-        L238-249: add appropriate references.

            Reference of FLICA regent added

 Minor comments

 All minor comments have been addressed and highlighted in yellow in the text